# Matrix Metalloproteinase-9 Enhances Osteoclastogenesis: Insights from Transgenic Rabbit Bone Marrow Models and In Vitro Studies

**DOI:** 10.3390/ijms26073194

**Published:** 2025-03-29

**Authors:** Yajie Chen, Jialun Zou, Manabu Niimi, Xuan Qiu, Shuang Zhang, Han Yang, Maobi Zhu, Jianglin Fan

**Affiliations:** 1Guangdong Province Key Laboratory, Southern China Institute of Large Animal Models for Biomedicine, School of Pharmacy and Food Engineering, Wuyi University, Jiangmen 529020, China; j002676@wyu.edu.cn (Y.C.); msc_jialun_zou@163.com (J.Z.); q613550481@foxmail.com (X.Q.); zs2804767433@163.com (S.Z.); yh102359@163.com (H.Y.); 2Department of Molecular Pathology, Interdisciplinary Graduate School of Medicine, University of Yamanashi, Chuo 409-3898, Japan; manabun@yamanashi.ac.jp

**Keywords:** osteoclast, MMP-9, bone marrow cells, inflammation, transgenic rabbit

## Abstract

Osteoclastogenesis is tightly regulated by receptor activator of nuclear factor kappa-B ligand (RANKL) signaling, yet the role of matrix metalloproteinase-9 (MMP-9) in this process remains controversial. We established a high-yield osteoclastogenesis system using cryopreserved rabbit bone marrow cells (1 × 10^9^ cells/femur) treated with Macrophage colony-stimulating factor (M-CSF) and RANKL. Bone marrow cells from MMP-9 transgenic rabbits (macrophage-specific overexpression) and MMP-9-transfected RAW264.7 macrophages were compared to wild-type controls. MMP-9 overexpression increased osteoclastogenesis 5.5-fold (20 ng/mL RANKL, * *p* < 0.01) while suppressing inflammatory cytokines (*IL-1β, TNF-α*). RAW264.7 macrophages stably transfected with human MMP-9 similarly exhibited reduced inflammatory cytokine levels and enhanced osteoclastogenesis. MMP-9 acts as a dual regulator of osteoclastogenesis and inflammation, suggesting therapeutic potential for osteoporosis management.

## 1. Introduction

Osteoclasts, the bone-resorbing cells derived from hematopoietic precursors, play a pivotal role in maintaining skeletal homeostasis through tightly regulated interactions with osteoblasts [1]. These multinucleated giant cells degrade mineralized bone matrix by acid secretion (via V-ATPase) and release of proteolytic enzymes, including metalloproteinases and cathepsin K [2]. They are crucial in bone physiological and pathophysiological processes, regulating bone resorption, remodeling, calcium homeostasis, and intercellular communication, thereby maintaining bone balance and skeletal health [2]. The process of osteoclastogenesis is strictly regulated by the receptor activator of nuclear factor kappa-B ligand (RANKL), primarily expressed by bone marrow mesenchymal cells, osteoblasts, and immune cells [1]. Dysregulated osteoclast activity underpins multiple skeletal pathologies, such as osteoporosis, osteoarthritis, and bone metastasis in cancer [3,4,5]. Current antiresorptive therapies, such as bisphosphonates and denosumab, primarily target osteoclast activity but are limited by side effects and their inability to address inflammatory drivers of bone loss. Thus, identifying novel regulators of osteoclastogenesis that combine anti-inflammatory and bone-protective effects remains an urgent need.

Matrix metalloproteinases (MMPs), a family of zinc-dependent endopeptidases, include several members highly expressed in bone and cartilage [6]. Recent reports have linked MMP mutations to skeletal diseases in humans [7]. MMP-9 (gelatinase B), which is uniquely secreted by osteoclasts into resorption lacunae to degrade type IV collagen and denatured collagens [8], shows elevated expression levels correlating with osteoarthritis severity [9]. While MMP-9 inhibitors (including tetracycline analogs) impair osteoclastogenesis in vitro [10], and *MMP-9* antisense treatment inhibited bone resorption through suppressing osteoclast migration [11], other studies demonstrated that RANKL stimulates osteoclastogenesis while increasing *MMP-9* expression [12]. However, whether MMP-9 overexpression directly influences osteoclastogenesis remains unclear.

Rabbits (*Oryctolagus cuniculus*) are important and widely used animal models in biomedical research, providing a translational bridge due to their anatomical similarity to human skeletal systems [13,14]. Unlike rodent models, they exhibit Haversian remodeling and their joints’ anatomy and trabecular bone architecture closely resemble those of humans, facilitating spontaneous osteoarthritis modeling [15,16]. Moreover, CT/MRI resolution enables longitudinal tracking of bone microstructure. Compared to mice, a single rabbit femur yields more than 100 times the number of bone marrow cells, enabling large-scale in vitro assays. With the development of gene editing technologies, rabbits have become a viable model for studying genetic factors in skeletal development and joint diseases. Leveraging these strengths, we generated transgenic rabbits overexpressing human MMP-9 under the control of macrophage-specific promoter [17]. In this study, we developed a simple and stable method for differentiating osteoclasts from rabbit bone marrow cells using M-CSF and RANKL induction. Furthermore, we investigated the role of MMP-9 in osteoclastogenesis using bone marrow cells from MMP-9 transgenic rabbits and RAW264.7 cell lines. Our findings establish MMP-9 as a therapeutic target with dual action in balancing bone resorption and inflammation, representing a paradigm shift in osteoporosis management.

## 2. Results

In the current study, up to 1 × 10^9^ cells were collected from both femurs of a rabbit, representing a 100-fold increase compared to the yield from a mouse. The large number of marrow cells obtained from a single animal provides greater homogeneity than samples pooled from multiple individuals, enhancing the reliability of subsequent in vitro studies. As shown in Figure 1A, after 5 days of M-CSF treatment, bone marrow-derived cells showed positive staining for the rabbit macrophage marker RAM11. Notably, cryopreserved bone marrow cells retained macrophage differentiation capabilities comparable to freshly isolated cells. Our results demonstrated that in the absence of RANKL stimulation, bone marrow cells cultured with M-CSF alone retained mononuclear morphology with TRAP-positive staining (Figure 1B, upper panel). Following the 5-day RANKL treatment, cells exhibited characteristic osteoclast morphology including multinucleation, distinct clear zones, and ruffled borders (Figure 1B). To optimize differentiation, we tested three RANKL concentrations in 24-well plates (14 mm diameter). The average numbers of mature osteoclasts were 78, 182, and 146 at 20, 50, and 70 ng/mL RANKL, respectively (Figure 1C). Intriguingly, increasing RANKL to 70 ng/mL demonstrated a suppressive trend in osteoclastogenesis.

Comparative analysis of bone marrow cells from transgenic and wild-type rabbits was conducted to determine whether MMP-9 overexpression affects macrophage differentiation. The immunofluorescence staining revealed that after 6 days of M-CSF culture, bone marrow cells from transgenic rabbits exhibited significantly elevated MMP-9 expression (68.4 ± 6.1 vs. 6.0 ± 2.2 in wild-type rabbits) (Figure 2A). Western blot analysis confirmed markedly higher MMP-9 levels in culture media from transgenic samples compared to wild-type groups. Notably, overexpressed MMP-9 did not alter macrophage differentiation efficiency (75.7 ± 6.4 vs. 75.6 ±10.9 in wild-type rabbits) (Figure 2B). Unexpectedly, MMP-9 overexpression showed anti-inflammation effects characterized by suppressed expression of pro-inflammatory factors *IL-1β*, *MCP-1*, and *TNF-α* (Figure 2C).

Next, we investigated the role of MMP-9 in osteoclastogenesis. Bone marrow cells from wild-type and transgenic rabbits were cultured with M-CSF and RNAKL as described. Results demonstrated that MMP-9 overexpression enhanced osteoclastogenesis (5.5-fold increase at 20 ng/mL RANKL and 3.2-fold increase at 50 ng/mL RANKL) (Figure 3). Furthermore, RANKL treatment upregulated MMP-9 expression (Figure 3A–C). To validate these findings, we generated R W264.7 cells with stable human MMP-9 overexpression. The lentiviral vector construct for MMP-9 overexpression is shown in Appendix A, where human MMP-9 cDNA and EGPF expression are driven by cytomegalovirus (CMV) promoter, with EGFP serving as a transfection marker. Western blot confirmed substantial MMP-9 expression in both cell lysis and culture media (Figure 4A). Notably, MMP-9 overexpression significantly suppressed the expression of proinflammatory genes *MCP-1* and *TNF-α* (Figure 4B), while promoting osteoclastogenesis (Figure 4D).

## 3. Discussion

The role of MMP-9 in osteoclastogenesis remains controversial. In our previous work, we generated transgenic rabbits with macrophage-specific overexpression of human MMP-9 via pronuclear microinjection technique in our lab [17]. Our results demonstrated that bone marrow cells from transgenic rabbits showed enhanced RANKL-induced osteoclastogenesis compared to wild-type controls. Typically, in vitro differentiated mature osteoclasts contain >20 nuclei with a diameter reaching 100 μm [18,19]. Our protocol enables stable and efficient differentiation of rabbit bone-marrow-derived osteoclasts containing 30–60 nuclei pre-cell. Notably, cryopreserved bone marrow cells retained osteoclastogenic potential.

In bone physiology, MMPs orchestrate bone remodeling, repair/regeneration, and pathological resorption in osteoporosis and bone metastasis [8]. As the most abundant MMP in osteoclasts [8], MMP-9 was previously shown to enhance cell migration and collagen degradation capacity [17]. Emerging evidence suggests that MMP-9 modulates bone angiogenesis, whereas its inhibition impairs osteoclast recruitment, leading to hypertrophic cartilage accumulation [11,20,21]. Intriguingly, a recent study revealed that endothelial-derived (rather than osteoclast-derived) MMP-9 drives cartilage resorption [22]. Furthermore, MMP-9 may act as an epigenetic regulator sustaining osteoclast activation [23]. Mechanistically, osteoblast-derived RANKL—the master regulator of osteoclastogenesis—induces MMP-9 expression through NFATc1-mediated transactivation of its promoter [12].

MMP-9 exerts context-dependent dual roles in inflammation, modulating both pro- and anti-inflammatory effects. Its substrates extend beyond denatured collagens to include epigenetic regulators; for instance, MMP-9-dependent proteolysis of histone H3 N-terminal tails is essential for activating gene transcription during osteoclastogenesis [10,24]. Additionally, MMP-9 processes pro-IL-1β into its active form and cleaves both mature IL-1β and membrane-bound TNF-α, thereby amplifying pro-inflammatory responses [25]. Intriguingly, our findings revealed that macrophage-specific MMP-9 overexpression exerts anti-inflammatory effects, consistent with reports showing that human MMP-9 overexpression in macrophages improves post-myocardial infarction cardiac function and attenuates age-related cardiac fibrosis in mice [26,27]. However, the mechanism underlying MMP-9-mediated inflammation suppression remains elusive.

Notably, compared to bone marrow-derived macrophages, peritoneal macrophages exhibit reduced osteoclastogenic potential [28], suggesting that macrophage polarization states influence their differentiation capacity. Under inflammatory conditions, osteoclast overactivation drives pathological bone loss. Paradoxically, glucocorticoids used to suppress chronic inflammation paradoxically increase osteoporosis risk [29], while tetracycline inhibits MMP-9 activity to ameliorate bone loss [10], highlighting the complex interplay between inflammation and osteoclastogenesis. In osteoarthritis, a chorionic inflammatory disease, elevated MMP-9 levels correlate with cartilage degradation and disease progression. Within the osteoarthritic microenvironment, osteoclasts, macrophages, and neutrophils constitute the primary MMP-9 sources [30]. And clinical studies showed suppressing IL-1β and TNF-α both have benefits for osteoarthritic management [31,32]. Targeted inhibition of MMP-9 demonstrates therapeutic potential by suppressing inflammation and preserving cartilage integrity.

While RAW264.7 remains the only established cell line capable of osteoclastogenesis, these cells exhibit reduced sensitivity and differentiation stability compared to bone marrow-derived macrophages during RANKL induction, as demonstrated by our and others’ studies [33]. Recent advances in CRISPR-Cas9 technology have enabled the generation of gene-knockout and point-mutation rabbit models [34], positioning genetically modified rabbit bone marrow cells as a valuable resource for translational osteoclast research. This study reveals that macrophage-specific overexpression of MMP-9 in transgenic rabbits enhances RANKL-induced osteoclastogenesis while suppressing pro-inflammatory cytokine production, establishing MMP-9 as a dual-function regulator of bone remodeling and inflammation. These findings highlight MMP-9’s therapeutic potential for osteoporosis management, particularly in reconciling pathological bone resorption with inflammatory cascades. Future investigations should assess the therapeutic potential of MMP-9 inhibitors in this preclinical model and explore their applicability in inflammatory bone disorders.

## 4. Materials and Methods

### 4.1. Isolation of Rabbit Bone Marrow Cells

Rabbits were housed in the facilities under controlled 20–24 °C with a humidity of around 40–70% with a 12:12 h light–dark cycle. After 3 months of age, the rabbit was kept in a single cage with ad libitum feeding and drinking. MMP-9 transgenic rabbits were generated in our lab [17]. Wild-type and same-litter MMP-9 Japanese White Rabbits were used in this study. Male rabbits aged 6–9 months old were euthanized by intravenous injection of tiletamine and zolazepam (3 mg/kg), followed by 10% KCl (1.5 mL/kg). Femurs were immediately separated and processed in a biosafety cabinet. The femur was immersed in 70% cold ethanol for disinfection, and then washed with PBS. The femur was fractured, and 5 mL of cold phosphate buffer salt (PBS) was used to aspirate and wash out the bone marrow. The bone marrow was disaggregated using a surgical blade and pipetting to obtain a cell suspension, which was filtered through a 40 μm cell strainer, and centrifuged at 2000 rpm for 5 min. The cell pellet was resuspended in 3 mL of red cell lysis buffer and incubated on ice for 3 min. After centrifugation, bone marrow cells were collected for further experiments or kept in cryopreserved condition in liquid nitrogen. All animal experiments were conducted with the approval of the Animal Ethics Committee of Yamanashi University and Wuyi University, and the ethical committee number is CN2021004.

### 4.2. In Vitro Osteoclasts Differentiation

Bone marrow cells were cultured in RPMI-1640 medium with 10% FBS in a 24-well plate at a density of 1 × 10^6^ cells per well with recombinant human M-CSF (PeproTech, Rocky Hill, NJ, USA) as 20 ng/mL. After 6 days, fresh medium was added with M-CSF and RANKL (PeproTech) and cultured for an additional 5 days. Cells were then fixed with 4% paraformaldehyde for subsequent analysis [35].

### 4.3. TRAP Staining

Tartrate-Resistant Acid Phosphatase (TRAP) Staining Fixed cells were washed with pre-cooled PBS and then incubated with a solution containing 0.1 mg/mL naphthol AS-MX (Sigma-Aldrich, St. Louis, MO, USA) and 0.08 mg/mL Fast Red Violet LB Salt (Sigma) in 8 mM sodium tartrate and 30 mM sodium acetate (pH 5.0) at 37 °C for 2 h. After washing with distilled water, the cells were mounted for observation. Mature osteoclasts were identified as TRAP-positive with more than three nuclei [17].

### 4.4. Immunoblotting Analysis

Cell culture supernatants were collected and centrifuged at 4 °C 3000× *g* for 15 min to remove cell debris. Denatured supernatant samples (20 μL) were separated by 10% SDS-PAGE, transferred to a nitrocellulose membrane, and incubated overnight at 4 °C with monoclonal antibody (Cat No. 32160702, Sigma-Aldrich, St. Louis, MO, USA) to detect MMP-9 [17]. 

### 4.5. Immunofluorescence Staining

Fixed cells were washed with PBS, permeabilized, and blocked with 0.1% Triton X-100 and 5% fetal bovine serum for 30 min at room temperature (RT). Cells were incubated with primary antibodies overnight at 4 °C. The following primary antibodies were used: mouse anti-MMP-9 (1:100), mouse anti-RAM11 (macrophage) (1:400), and rabbit-anti-β actin (1:200). Alexa Fluor 488/568-conjugated secondary Abs (#A21202/A10042, Thermo Fisher Scientific, Carlsbad, CA, USA) were used at a dilution of 1:200 at RT for 1 h. Nuclei were counterstained with DAPI (#62248, Thermo Fisher Scientific). Images were captured using a fluorescent microscope (Nikon, ECLIPSE Ti2, Tokyo, Japan) [36]. And the colocalization of MMP-9 or RAM11 with DAPI was quantified based on ImageJ (https://github.com/cberri/cFOS_ManualAnnotations_ImageJ-Fiji, accessed on February 2023).

### 4.6. Quantitative Real-Time Reverse Transcription Polymerase Chain Reaction (qRT-PCR)

Total RNA was extracted using TRIzol reagent (Cat# 15596026, Thermo Fisher Scientific, Waltham, MA, USA). RNA purity was verified by A260/A280 ratios between 1.9 and 2.0. First-strand cDNA synthesis was performed with the iScript cDNA Synthesis Kit (Bio-Rad, Hercules, CA, USA) according to the manufacturer’s protocol. Quantitative real-time PCR (qRT-PCR) was conducted using 2X SYBR Premix (Bio-Rad) on a Roche LightCycler 96 system, with thermal cycling parameters as follows: 95 °C for 120 s (initial denaturation), followed by 40 cycles of 95 °C for 5 s, 60 °C for 10 s, and 72 °C for 30 s. GAPDH served as the endogenous control, and relative gene expression was calculated using the 2^−ΔΔCt^ method. Primer sequences are listed in Table 1.

### 4.7. RAW264.7 Cell-Line Culture

The RAW264.7 cell line was purchased from Procell Life Science & Technology (Wuhan, China) and maintained in a high-glucose DMEM medium (DMEM, HyClone, Logan, UT, USA) under a 5% CO_2_ atmosphere with 95% humidity at 37 °C. Culture media were supplemented with 10% fetal bovine serum and 1% penicillin/streptomycin (100 U/mL penicillin, 100 μg/mL streptomycin). Cells were passaged upon reaching 80–90% confluence. For osteoclast induction, cells were seeded into 96-well plates in a complete medium containing RANKL (20 ng/mL). After 5 days, cells were fixed with 4% paraformaldehyde for TRAP staining and immunofluorescence analysis.

### 4.8. Lentivirus-Mediated Expression of MMP-9

The human MMP-9 coding sequence (CDS) cloning vector was provided by Wuhan Miaoling Biologicals (Wuhan Miaoling Biologicals, China). Recombinant lentivirus was packaged by Cyagen Biosciences (Cyagen Biosciences, Guangzhou, China). EGFP served as a reporter gene, with puromycin selection used to enrich transfected cells. Puromycin treatment achieved 100% EGFP-positive cells in both control and MMP-9-transfected groups. The vector construct is detailed in Appendix A.

### 4.9. Statistical Analysis

Data are presented as mean ± standard deviation (SD). All data were first assessed for normality (Shapiro–Wilk test). A student’s *t*-test or Wilcoxon rank-sum test was employed to compare differences between the two groups. One-way ANOVA test, followed by all pairwise comparisons, was used to determine differences among the three groups. Statistical analysis was performed using SPSS 16.0 software. A *p*-value of less than 0.05 was considered statistically significant. Graphs were generated using Prism 9.5 software (GraphPad Software, San Diego, CA, USA).

## Figures and Tables

**Figure 1 ijms-26-03194-f001:**
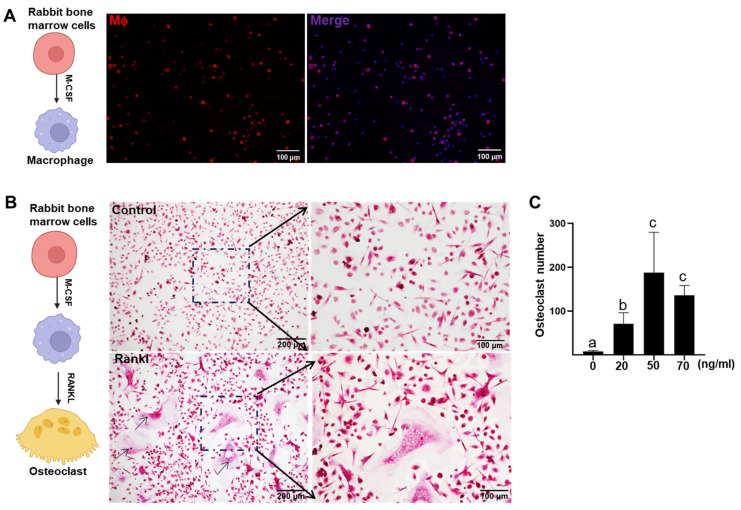
In vitro osteoclastogenesis from rabbit bone marrow cells: (**A**) Schematic illustration of experimental design (left panel). Representative images of bone marrow-derived macrophages immunohistochemically stained with the anti-rabbit macrophage marker RAM11 (red), with nuclei counterstained with DAPI (blue), and merged results are shown. (**B**) Schematic illustration of experimental design (left panel). Representative image of mature osteoclasts demonstrating characteristic features including multinucleation, ruffled borders, and TRAP-positive staining. Arrow points matured osteoclasts. (**C**) Quantification of osteoclast numbers in 24-well plates. Media were supplemented with RANKL at 0, 20, 50, or 70 ng/mL. Data are presented as mean ± SD (N = 4). Statistical significance was assessed by one-way ANOVA, with different lowercase letters (a, b, c) indicating significant differences between groups (*p* < 0.05).

**Figure 2 ijms-26-03194-f002:**
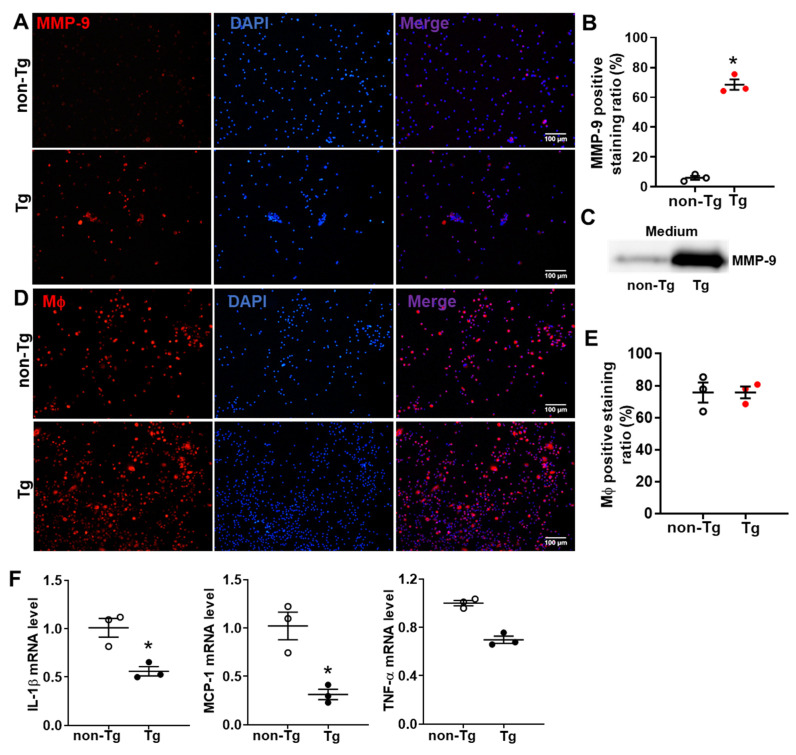
Characteristics of rabbit bone marrow cell-derived macrophage: (**A**) Representative immunofluorescence image showing MMP-9 expression (red) in bone marrow cell-derived macrophage from transgenic and wild-type rabbits with nuclear counterstained with DAPI (blue) and merged channels displayed. All images were the same magnification. (**B**) Quantification of MMP-9 positive cell proportions. (**C**) Immunoblot analysis of MMP-9 protein levels in culture supernatant medium after 6 days of M-CSF treatment in both groups. (**D**) Representative immunofluorescence image of macrophage marker RAM11 (red) expression, with nuclear stained by DAPI (blue) and merged channels shown. (**E**) Quantification of RAM11-positive cell proportions. (**F**) qRT-PCR analysis of inflammatory gene mRNA levels (IL-1β, MCP-1, TNF-α). Data are presented as mean ± SD from three independent experiments. Statistical significance was determined by Student’s *t*-test, with asterisks (*) indicating a significant difference (*p* < 0.05).

**Figure 3 ijms-26-03194-f003:**
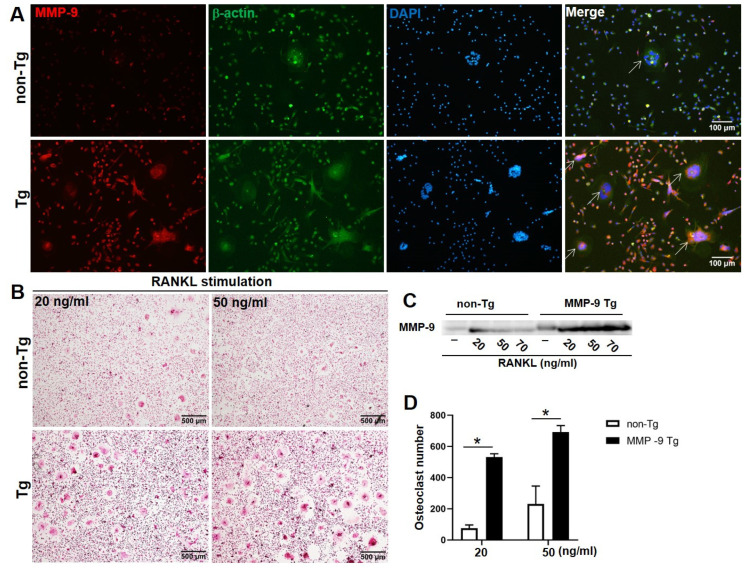
MMP-9 overexpression enhances osteoclastogenesis: (**A**) Representative immunofluorescence image of MMP-9 (red) and α-actin (green) in monocytes and osteoclasts derived from bone marrow cells of MMP-9 transgenic or wild-type rabbits. Arrows indicate multinuclear cells and all images were the same magnification. (**B**) TRAP staining of osteoclasts differentiated from the bone marrow cells with RANKL stimulation (20 and 50 ng/mL). (**C**) Immunoblot analysis of MMP-9 expression in culture supernatants from MMP-9 transgenic and non-transgenic rabbits under different RANKL concentrations. (**D**) Quantification of osteoclast numbers in a 24-well plate with RANKL at 20 or 70 ng/mL. Data are mean ± SD (N = 4). Statistical significance was determined by Student’s *t*-test; asterisks (*) indicate significant differences (*p* < 0.05).

**Figure 4 ijms-26-03194-f004:**
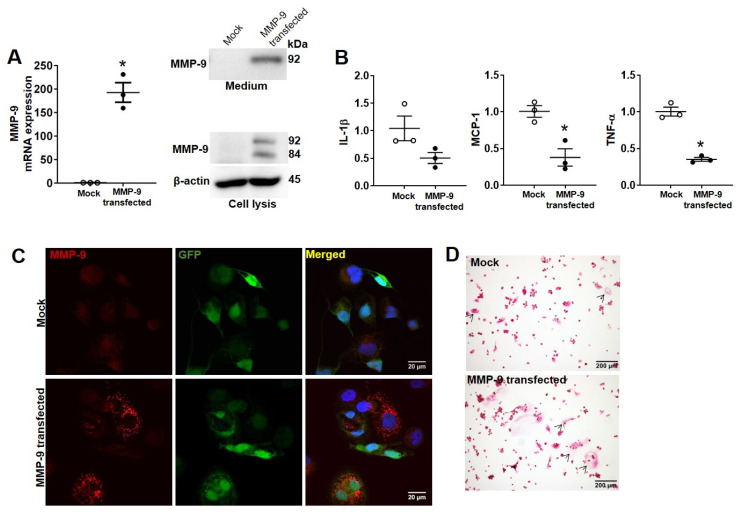
Stable overexpression of human MMP-9 in RAW264.7 cells promotes osteoclastogenesis: (**A**) Human MMP-9 expression was verified by qRT-PCR (left panel) and Western blot analysis of MMP-9 level in culture supernatants and cell lysates (right panel). (**B**) qRT-PCR analysis of anti-inflammatory gene mRNA levels. Data are presented as mean ± SD from three independent experiments. Statistical significance was determined by Student’s *t*-test; asterisks (*) indicate significant differences (*p* < 0.05). (**C**) Representative immunofluorescence images of MMP-9 (red), GPF (green), and DAPI (blue) in mock-transfected and MMP-9-transfected cells. (**D**) TRAP staining of osteoclasts differentiated from RAW264.7 cells stimulated with RANKL (20 ng/mL). Arrowheads denote multinuclear cells.

**Table 1 ijms-26-03194-t001:** Primer sequence used in reverse transcription-quantitative polymerase chain reaction.

Genes	Primer Sequences (5′–3′)	
	Forward	Reverse
Mouse GAPDH	AGGTCGGTGTGAACGGATTTG	GGGGTCGTTGATGGCAACA
Mcie IL-1b	TTCAGGCAGGCAGTATCACTC	GAAGGTCCACGGGAAAGACAC
Mcie TNF-a	ACGGCATGGATCTCAAAGAC	AGATAGCAAATCGGCTGACG
Mcie MCP-1	CTTCTGGGCCTGCTGTTCA	CCAGCCTACTCATTGGGATCA
Rabbit GAPDH	ATCACTGCCACCCAGAAGAC	GTGAGTTTCCCGTTCAGCTC
Rabbit IL-1b	GGAGAGCTCTTTCCCACCAG	TGGGTACCAAGGTTCTTTGAA
Rabbit TNF-a	ATGGTCACCCTCAGATCAGC	CTGGTTGTCCGTGAGCTTC
Rabbit MCP-1	AGCACCAAGTGTCCCAAAGA	TGTGTTCTTGGGTTGTGGAA

## Data Availability

Data are available on request due to restrictions. The data presented in this study are available on request from the corresponding author.

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
