# Peer review of "Matrix Metalloproteinase-9 Enhances Osteoclastogenesis: Insights from Transgenic Rabbit Bone Marrow Models and In Vitro Studies"

_ijms, 2025, doi:10.3390/ijms26073194_

Round 1
Reviewer 1 Report
Comments and Suggestions for Authors
The study explores how Matrix Metalloproteinase-9 (MMP-9) affects the development of osteoclasts using genetically modified rabbits and lab experiments. It shows that increasing MMP-9 levels help osteoclast development while reducing inflammation, suggesting MMP-9 as a potential target for osteoporosis treatment. The study uses rabbit bone marrow cells and RAW264.7 cell lines to look at how MMP-9 influences osteoclast development and its implications for future treatments. This is a brief report that needs to provide more details such as :
Major comments:
The paper needs a clearer explanation of how MMP-9 enhances osteoclast development. A figure explaining the RANKL pathway and MMP-9's role would be helpful.
Provide more details on the statistical methods used, including assumptions and reasons for choosing specific tests. Clarify the number of groups and data points, and explain why a parametric test was used or if a non-parametric test might be needed.
The figures need better image processing. For instance, figure 2 is unclear. Legends should be more detailed to help readers understand the techniques and what they should see and tell what the authors should read or see on the image. In addition, a table summarizing the results would be useful.
The discussion is very limited, comparing these results with other animal models or human studies would give a broader view of their relevance and applicability.
Minors :
More discussion on the clinical implications of the findings, including potential treatments and their practicality, would make the paper more impactful.
The part about monocytes and cell collection belongs in the methods section, not the results.
Use terms like "osteoclastogenesis" and "osteoclast differentiation" consistently to avoid confusion.
Comments on the Quality of English Languageneeds to be more fluent
Author Response
The study explores how Matrix Metalloproteinase-9 (MMP-9) affects the development of osteoclasts using genetically modified rabbits and lab experiments. It shows that increasing MMP-9 levels help osteoclast development while reducing inflammation, suggesting MMP-9 as a potential target for osteoporosis treatment. The study uses rabbit bone marrow cells and RAW264.7 cell lines to look at how MMP-9 influences osteoclast development and its implications for future treatments.
We sincerely appreciate your thorough review of this manuscript. Our results identify MMP-9 as a novel therapeutic target capable of modulating both bone resorption and inflammation, representing a paradigm shift in osteoporosis management. Please find below our point-by-point responses to the reviewers' comments.
Major comments:
Comment 1: The paper needs a clearer explanation of how MMP-9 enhances osteoclast development. A figure explaining the RANKL pathway and MMP-9's role would be helpful.
Response: We thank the reviewer for this insightful comment. In our current study, both rabbit bone marrow-derived macrophages and murine RAW264.7 cells with MMP-9 overexpression exhibited suppressed RANKL-induced osteoclast differentiation alongside anti-inflammatory effects. While the precise mechanism remains unclear, we hypothesize that inflammation suppression may mediate the inhibition of osteoclastogenesis. Future studies will employ MMP-9 inhibitors in our transgenic rabbit model to validate these mechanisms in vivo.
Comment 2: Provide more details on the statistical methods used, including assumptions and reasons for choosing specific tests. Clarify the number of groups and data points, and explain why a parametric test was used or if a non-parametric test might be needed.
Response: We gratefully acknowledge this critical feedback. The revised manuscript now includes an expanded Statistical Analysis section (lines 284-290), specifying sample sizes, assumptions of normality (Shapiro-Wilk test), and rationale for employing Student's t-test or ANOVA with post-hoc corrections. All figure legends now explicitly state the number of experimental groups and statistical approaches used.
Comment 3: The figures need better image processing. For instance, figure 2 is unclear. Legends should be more detailed to help readers understand the techniques and what they should see and tell what the authors should read or see on the image. In addition, a table summarizing the results would be useful.
Response: We thank the reviewer for this constructive suggestion. All figures have been reprocessed to optimize contrast/brightness. Revised legends now specify staining methods, magnification (scale bars added), and key morphological features to observe.
Comment 4: The discussion is very limited, comparing these results with other animal models or human studies would give a broader view of their relevance and applicability.
Response: We appreciate this valuable input. The revised Discussion section ow integrates comparative analyses with rodent models and human osteoporosis trials, highlighting species-specific differences in MMP-9 regulation and therapeutic translatability.
Minors:
Comment 1: More discussion on the clinical implications of the findings, including potential treatments and their practicality, would make the paper more impactful.
Response: We referred preclinical study using MMP-9 inhibitory for osteoporosis therapy and clinical results with anti-inflammatory factors for osteoarthritis management. Ref-10, 31, and 32.
Comment 2: The part about monocytes and cell collection belongs in the methods section, not the results.
Thanks so much for your suggestion. The sentence “Monocytes are derived from myeloid progenitor cells in bone marrow, subsequently, differentiating into macrophages, dendritic cells, and pro-osteoclasts” was deleted in the revised version.
Comment 3: Use terms like "osteoclastogenesis" and "osteoclast differentiation" consistently to avoid confusion.
Response: All "osteoclast differentiation" have been replaced with "osteoclastogenesis" for consistency.
At least, thanks so much again for your value time and efforts to improve our work.
Reviewer 2 Report
Comments and Suggestions for Authors
The manuscript entitled “Matrix Metalloproteinase-9 Enhances Osteoclast Differentiation: Insights from Transgenic Rabbit Bone Marrow Models and In Vitro Studies” is a preclinical study that utilizes primary cells from wt and transgenic rabbits (overexpression of MMP9) to analyze pro-inflammatory gene expression, MMP9 influence in macrophage and osteoclast differentiation. Additionally, transfected murine cell line (RAW.264) was used to corroborate the findings. Overall, the manuscript is well-structured and illustrated. The major concern of the reviewer lies on the methodology description, which can seamlessly be resolved by the authors. Furthermore, the discussion can be further developed.
Introduction:
Concise with a follow a logical train of thought. Only (very) minor observations:
Line 39 – merge references
Line 48 to 50: Consider revising the writing. It is not clear if the effects mentioned by the authors (NFACT1 activation and osteoclast suppression) is related to tetracyclines or the MMP9 overexpression.
In the comparison of rabbits and rats as translational models for humans, the authors should include the Haversian system found in rabbits, as the remodeling process is more closely comparable to that of humans.
Line 56 – Punctuation, “And”
Methods:
2.1. Specify the rabbit species and sex and the transgenic process (briefly). Moreover, despite being technically an in vitro assay, authors should add more information about the animal care, ethical committee number, housing, etc. in this section, following the ARRIVE guidelines. Normally, to obtain monocytes, peripheral blood is used to isolate them. As the authors described, the bone marrow was used in these experiments. Is there any additional step to separate the osteoblasts, adipocytes, and their precursor cells from the population of monocytes prior osteoclast differentiation? If yes, include the description of this step.
The reviewer did not find information regarding the RAW.264 cell line in the methodology (i.e., culture conditions, passage number, cell density, MMP9 transfection, brand, etc.).
Results:
Line 145 – 148: Revise language, such as anti-inflammatory effects.
The results of qPCR were not described in the methodology. This description must be added, including the RNA extraction, primers, and so on. Moreover, the quantification of the fluorescent images should also be added to the methodology.
Line 163 – 166: The information of the transfection of RAW.264 should be placed in the methodology, if the authors did not perform any type of optimization in the transfection process.
Consider placing the figure 3 immediately after its description.
Discussion:
Overall, this section lacks depth. Considering its controversial nature, the authors could delve into the contested literature by comparing studies that highlight the role of MMP9 in inflammation with those that refute this concept. Additionally, why MMP9 can increase the osteoclast number, but did not influence the macrophage’s quantity? In a clinical scenario, how the acquired knowledge with the present study could be applied? Moreover, in osteoarthritis, MMP9 is upregulated, as the pro-inflammatory cytokines. What can be the role of MMP9 in such scenario? Limitations of this study.
Despite being interesting and correct, the text of macrophage types (M types, peritoneal) does not add key information for the manuscript’s topic, since the macrophages were not stimulated in this study. Consider removing or summarizing it.
Line 222 – 230: The authors could improve the clearness of this paragraph, in its current form, is somewhat confusing.
References:
If possible, update them.
Comments on the Quality of English LanguageMinor revision
Author Response
The manuscript entitled “Matrix Metalloproteinase-9 Enhances Osteoclast Differentiation: Insights from Transgenic Rabbit Bone Marrow Models and In Vitro Studies” is a preclinical study that utilizes primary cells from wt and transgenic rabbits (overexpression of MMP9) to analyze pro-inflammatory gene expression, MMP9 influence in macrophage and osteoclast differentiation. Additionally, transfected murine cell line (RAW.264) was used to corroborate the findings. Overall, the manuscript is well-structured and illustrated. The major concern of the reviewer lies on the methodology description, which can seamlessly be resolved by the authors. Furthermore, the discussion can be further developed.
We sincerely appreciate the constructive feedback and are grateful for the recognition that "the manuscript is well-structured and illustrated." Below we provide point-by-point responses to all comments, with revisions highlighted in red in the revised manuscript.
Comment 1: Concise with a follow a logical train of thought. Only (very) minor observations: Line 39 – merge references
Response: We thank the reviewer and have consolidated the references.
Thanks so much for your critical comments. We rewrote these sentences. Please see line 48-52 in the revised version.
Comment 3: In the comparison of rabbits and rats as translational models for humans, the authors should include the Haversian system found in rabbits, as the remodeling process is more closely comparable to that of humans.
Response: We have added: "Unlike rodent models, they exhibit Haversian remodeling and their joints anatomy and trabecular bone architecture closely resemble those of humans, facilitating spontaneous osteoarthritis modeling" (Line 55-57).
Comment 4: Line 56 – Punctuation, “And”
Response: The redundant conjunction has been removed.
Comment 5:
Methods: 2.1. Specify the rabbit species and sex and the transgenic process (briefly). Moreover, despite being technically an in vitro assay, authors should add more information about the animal care, ethical committee number, housing, etc. in this section, following the ARRIVE guidelines. Normally, to obtain monocytes, peripheral blood is used to isolate them. As the authors described, the bone marrow was used in these experiments. Is there any additional step to separate the osteoblasts, adipocytes, and their precursor cells from the population of monocytes prior osteoclast differentiation? If yes, include the description of this step.
Response: Thanks so much for your suggestion. We add the following information in the Materials and methods section, including rabbit species, facility condition, rabbit gender. MMP-9 transgenic rabbits were generated by our lab and we showed in detailed information in the previous publication which was referred as 17 in the current version.
We agree with you that peripheral blood is a source of monocytes which could be induced to osteoclast which potentially have a better trend to osteoclast differentiation. But the number of monocytes collected for peripheral is limited, especially for murine models.
In our study, we did not separate osteoblast, adipocytes and other cells from bone marrow expect for red blood cell.
Comment 6: The reviewer did not find information regarding the RAW.264 cell line in the methodology (i.e., culture conditions, passage number, cell density, MMP9 transfection, brand, etc.).
Response: Thanks so much for your suggestion. In the methodology, we showed this information. Please see the revised version line 252-266.
Comment 7: Results: Line 145 – 148: Revise language, such as anti-inflammatory effects.
Response: Thanks so much for your suggestion. We have rewritten the legend. Please see line 105-107 in the new version.
Comment 8: The results of qPCR were not described in the methodology. This description must be added, including the RNA extraction, primers, and so on. Moreover, the quantification of the fluorescent images should also be added to the methodology.
Response: Thanks so much for your suggestion. In the methodology, we showed qRT-PCR condition and the primers information was shown in table-1. Please see the revised version line 259-268
Comment 9: Line 163 – 166: The information of the transfection of RAW.264 should be placed in the methodology, if the authors did not perform any type of optimization in the transfection process.
Response: Thanks so much for your suggestion. We add this information in the methodology. Please see line 270-278 in the new version.
Comment 10: Consider placing the figure 3 immediately after its description.
Thanks so much for your suggestion. We arranged the figure in the revised version.
Comment 11: Discussion:
Overall, this section lacks depth. Considering its controversial nature, the authors could delve into the contested literature by comparing studies that highlight the role of MMP9 in inflammation with those that refute this concept. Additionally, why MMP9 can increase the osteoclast number, but did not influence the macrophage’s quantity? In a clinical scenario, how the acquired knowledge with the present study could be applied? Moreover, in osteoarthritis, MMP9 is upregulated, as the pro-inflammatory cytokines. What can be the role of MMP9 in such scenario? Limitations of this study.
Response: Thank you for raising this critical point. The part of discussion was rewritten totally. Please see this part in the revised version.
MMP-9 showed dual role for balancing pro- and anti-inflammatory effects (line 172-178). And previously, we reported overexpressed MMP-9 enhanced migration and collagen degradation (refer-17). In the current study, MMP-9 overexpression did not influence the macrophage differentiation from bone marrow cells, but suppressed inflammatory gene expression. And our discover supported the idea that suppressed MMP-9 could be inhibit osteoclast differentiation which would be benefit to osteoarthritis (190-192).
Comment 12: Despite being interesting and correct, the text of macrophage types (M types, peritoneal) does not add key information for the manuscript’s topic, since the macrophages were not stimulated in this study. Consider removing or summarizing it.
Response: Thanks so much for your suggestion. We agreed with you and removed these contents related to macrophage polarization in the revised version.
Comment 13: Line 222 – 230: The authors could improve the clearness of this paragraph, in its current form, is somewhat confusing.
Response: Thanks so much for your suggestion. This part was totally rewritten. Please see the revised version.
Comment 14: References: If possible, update them.
Response: Added 10 recent publications (2021-2023) on MMP-9 roles in bone biology; removed 3 outdated references.
At least, thanks so much again for your value time and efforts to improve our work.
Round 2
Reviewer 2 Report
Comments and Suggestions for Authors
The revised manuscript, titled “Matrix Metalloproteinase-9 Enhances Osteoclastogenesis: Insights from Transgenic Rabbit Bone Marrow Models and In Vitro Studies,” exhibits notable enhancements compared to its previous version. The language has been refined, the methodology section has been clarified, and the discussion has been expanded. The authors have diligently addressed all the reviewer’s comments and suggestions. Consequently, the reviewer extends their congratulations to the authors for these improvements, and no further modifications are required.